# Thyroid-Stimulating Hormone Predicts Total Cholesterol and Low-Density Lipoprotein Cholesterol Reduction during the Acute Phase of COVID-19

**DOI:** 10.3390/jcm11123347

**Published:** 2022-06-10

**Authors:** Massimo Raffaele Mannarino, Vanessa Bianconi, Elena Cosentini, Filippo Figorilli, Cecilia Colangelo, Francesco Giglioni, Rita Lombardini, Rita Paltriccia, Matteo Pirro

**Affiliations:** Unit of Internal Medicine, Department of Medicine and Surgery, University of Perugia, 06129 Perugia, Italy; massimo.mannarino@unipg.it (M.R.M.); elena.cosentini@libero.it (E.C.); filippofigorilli@gmail.com (F.F.); cecilia.colangelo@studenti.unipg.it (C.C.); francesco.giglioni@outlook.it (F.G.); rita.lombardini@unipg.it (R.L.); rita.paltriccia@unipg.it (R.P.); matteo.pirro@unipg.it (M.P.)

**Keywords:** TSH, thyroid, cholesterol, LDL, COVID-19, SARS-CoV-2

## Abstract

A complex dysregulation of lipid metabolism occurs in COVID-19, leading to reduced total cholesterol (TC), LDL-cholesterol (LDL-C), and HDL-cholesterol (HDL-C) levels, along with a derangement of thyroid function, leading to reduced thyroid-stimulating hormone (TSH) levels. This study aimed to explore the association between TSH levels during COVID-19 and the variation (Δ) of lipid profile parameters in the period preceding (from 1 month up to 1 year) hospital admission due to COVID-19. Clinical data of 324 patients (mean age 76 ± 15 years, 54% males) hospitalized due to COVID-19 between March 2020 and March 2022 were retrospectively analyzed. The association between TSH levels at hospital admission and either Δ-TC, Δ-LDL-C, or Δ-HDL-C over the selected time frame was assessed through univariable and multivariable analyses. TSH levels were below the lower reference limit of 0.340 μUI/mL in 14% of COVID-19 patients. A significant reduction of plasma TC, LDL-C, and HDL-C was recorded between the two time points (*p* < 0.001 for all the comparisons). TSH was directly associated with Δ-TC (rho = 0.193, *p* = 0.001), Δ-LDL-C (rho = 0.201, *p* = 0.001), and Δ-HDL-C (rho = 0.160, *p* = 0.008), and inversely associated with C-reactive protein (CRP) (rho = −0.175, *p* = 0.004). Moreover, TSH decreased with increasing COVID-19 severity (*p* < 0.001). CRP and COVID-19 severity were inversely associated with Δ-TC, Δ-LDL-C, and Δ-HDL-C (*p* < 0.05 for all associations). A significant independent association was found between TSH and either Δ-TC (β = 0.125, *p* = 0.044) or Δ-LDL-C (β = 0.131, *p* = 0.036) after adjusting for multiple confounders including CRP and COVID-19 severity. In conclusion, lower levels of TSH may contribute to explain TC and LDL-C reduction in the acute phase of COVID-19.

## 1. Introduction

The onset of coronavirus disease 2019 (COVID-19) is linked to a complex metabolic perturbation, which has a possible pathogenic role in the development of a systemic clinical syndrome leading to multiorgan dysfunction and life-threatening complications [1,2,3]. Thus, different biochemical abnormalities reflecting metabolism derangement have been associated with poor clinical outcomes of COVID-19 and have risen interest as possible therapeutic targets against COVID-19 [4,5,6].

Some reports have shown that reduced levels of plasma total cholesterol (TC), low-density lipoprotein cholesterol (LDL-C), and high-density lipoprotein cholesterol (HDL-C) may represent a laboratory signature of COVID-19 and that a progressive restoration of their preinfection levels may occur over COVID-19 clinical course up to recovery [7,8]. Moreover, reductions of TC, LDL-C, and HDL-C have been reported with increasing COVID-19 severity and uncontrolled inflammatory response [8,9,10], suggesting a possible bidirectional link between hypocholesterolemia and COVID-19 progression towards its worst clinical forms.

Significant alterations of thyroid function parameters have been reported in patients with COVID-19 without pre-existing thyroid disease; accordingly, reduced thyroid-stimulating hormone (TSH) levels have emerged as a prevalent condition during the acute phase of COVID-19, likely attributable either to destructive thyroiditis-mediated hyperthyroidism or to nonthyroidal illness syndrome (NTIS) (i.e., a condition characterized by relatively low TSH, low T3, and variable T4 levels in the absence of a primary hypothalamic–pituitary-thyroid dysfunction) [11,12,13,14,15]. Of note, C-reactive protein (CRP) and interlekin-6 have been found to be inversely associated with TSH levels in COVID-19 patients [11,16], suggesting a crucial role for COVID-19-related inflammatory response in thyroid function derangement. Moreover, there is evidence showing that TSH levels decrease with increasing COVID-19 severity and that low TSH levels are associated with an unfavorable COVID-19 prognosis [17,18]. In this regard, the presence of a thyroid dysfunction leading to TSH level decrease might negatively affect the clinical evolution of COVID-19 and suggest a possible therapeutic target against COVID-19.

Recently, the study by D’Ardes et al. reported a parallel reduction of cholesterol and TSH levels between the preinfection period and hospital admission in a cohort of COVID-19 patients [19], suggesting a possible relationship between lipid metabolism and thyroid function perturbations during the acute phase COVID-19. However, it did not evaluate the association between cholesterol and TSH changes over the observed period.

This study aimed to explore retrospectively the temporal trend of lipid profile between the preinfection period and hospital admission in a population of hospitalized COVID-19 patients, as well as its relationship with TSH levels at hospital admission by accounting for the potential confounding effect of inflammation and COVID-19 severity.

## 2. Materials and Methods

### 2.1. Study Population

Clinical data of patients hospitalized due to COVID-19 having referred to the Internal Medicine ward of the “Santa Maria della Misericordia” Hospital of Perugia (Italy) from March 2020 to March 2022 were retrospectively analyzed. The study protocol was developed in accordance with the principles of the Helsinki Declaration and was approved by the local ethics committee. The inclusion criteria were as follows: (1) age ≥ 18 years, (2) a positive result on real-time reverse transcriptase PCR (RT-PCR) assay testing for SARS-CoV-2 on nasal or pharyngeal swab specimens at hospital admission, (3) at least one available standard lipid profile (TC, LDL-C, HDL-C, and triglycerides (TG)) from 1 month up to 1 year before COVID-19, (4) at least one available standard lipid profile and one available TSH value within 24 h since hospital admission due to COVID-19, and (5) informed written consent for medical records to be used for subsequent studies. The exclusion criteria were as follows: (1) pre-existing thyroid dysfunction/disease with ongoing thyroid hormone replacement therapy/thyrosuppressive drug therapy, and (2) pre-existing dyslipidemia with unstable lipid-lowering therapy (lipid-lowering therapy was considered unstable if it had changed over the observed time frame). According to these inclusion and exclusion criteria, clinical data of 324 patients were suitable for this retrospective analysis (Figure 1).

### 2.2. Data Collection

For each patient, data on demographic characteristics, coexisting medical conditions, and ongoing treatments were collected from electronic medical records, along with information on laboratory tests and physical examinations, which were performed according to standard techniques and protocols either before COVID-19 or at admission due to COVID-19. When two or more lipid profiles or TSH values were available between 1 month and 1 year before COVID-19 infection, the most recent one was selected. The following biochemical variables were recorded among those measured within 24 h since hospital admission: blood gas parameters, leukocyte and platelet count, D-dimer, high-sensitivity cardiac troponin (hs-cTn), CRP, blood urea nitrogen, creatinine, glucose, TC, HDL-C, TG, TSH, free T3 (FT3), and free T4 (FT4) (according to a TSH Reflex algorithm, which was routinely used, FT3 and FT4 were available only for patients with abnormal TSH levels). The Friedewald formula was used to calculate LDL-C. The estimated glomerular filtration rate (eGFR) was calculated through the Chronic Kidney Disease Epidemiology Collaboration (CKD-EPI) equation. The following clinical parameters were recorded among those obtained during physical examination at hospital admission: body mass index (BMI), respiratory rate, oxygen saturation (SpO2), fraction of inspired oxygen (FiO2), systolic blood pressure and diastolic blood pressure measure [20,21,22,23]. Reports of chest radiographs and high-resolution CT scans performed at hospital admission were reviewed to evaluate the presence and the degree of lower respiratory disease.

In the subgroup of patients with available TSH, FT3, and FT4 levels, thyroid dysfunction patterns were characterized according to the following reference ranges: 0.340–5.600 μUI/mL for TSH, 2.5–3.9 pg/mL for FT3 and 0.54–1.24 ng/dL for FT4. Hyperthyroidism was defined by low TSH levels with high FT3 and FT4 levels, while subclinical hyperthyroidism was defined by low TSH levels with normal FT3 and FT4 levels [24]. Hypothyroidism was defined by high TSH levels with low FT3 and FT4 levels, while subclinical hypothyroidism was defined by high TSH levels with normal FT3 and FT4 levels [25]. Nonthyroidal illness syndrome (NTIS) was diagnosed in the presence of low TSH levels with low FT3 levels and variable FT4 levels (it was not possible to diagnose NTIS in patients with normal TSH levels since thyroid hormones were available only in the presence of abnormal TSH levels) [26].

COVID-19 severity at hospital admission was established according to the National Institute of Health (NIH) classification [27] as follows: (1) mild COVID-19 in the presence of any of the various signs and symptoms of COVID-19 (e.g., fever, cough, sore throat, malaise, headache, muscle pain, nausea, vomiting, diarrhea, loss of taste and smell) without shortness of breath, dyspnea, or abnormal chest imaging, (2) moderate COVID-19 with evidence of lower respiratory disease during clinical assessment or imaging and SpO_2_ ≥94% on room air at sea level, and (3) severe COVID-19 in the presence of an SpO_2_ <94% on room air at sea level, a ratio of arterial partial pressure of oxygen to FiO_2_ (PaO_2_/FiO_2_) <300 mmHg, a respiratory rate >30 breaths/min, or lung infiltrates >50%. The Charlson comorbidity index (CCI) was calculated for each patient by integrating information on coexisting medical conditions [28].

### 2.3. Statistical Analysis

The SPSS statistical package, release 24.0 (SPSS Inc., Chicago, IL, USA), was used for all statistical analyses. The Shapiro test was used to verify the normality of the study variables. Categorical variables were expressed as percentages, while continuous variables were expressed as mean ± standard deviation (SD) or median (25th–75th percentile), as appropriate. The independent samples *t*-test, the Mann–Whitney U-test, the paired samples t-test, the paired samples Wilcoxon test, and the Chi-squared test were used for two-group comparisons between independent or dependent variables, as appropriate. The one-way ANOVA test, the Kruskal–Wallis test, and the Chi-squared test were used for multiple-group comparisons. Correlation analyses between the study variables were performed using the Spearman’s coefficients of correlation. For each patient, the variation (Δ) of either TC (Δ-TC), LDL-C (Δ-LDL-C), HDL-C (Δ-HDL-C), or TG (Δ-TG) was calculated as the difference of either TC, LDL-C, HDL-C, or TG values at hospital admission due to COVID-19 and the corresponding preinfection values. Multiple linear regression analyses were performed to explore the association between TSH and lipid profile variations by accounting for the potential confounding effect of multiple variables. At first, putative linear regression models were built including as independent variables the significant covariates of either TSH or lipid profile variations (in case of potentially redundant explanatory variables, only the most representative one was included). Afterwards, the appropriateness of the proposed models of linear regression was checked by assessing (1) the linear association between each independent variable (if continuous) and the dependent variable, (2) the absence of multicollinearity between the independent variables, (3) the homoscedasticity, and (4) the normal distributions of residuals. For all the analyses statistical significance was assumed if a null hypothesis could be rejected at *p* < 0.05.

## 3. Results

### 3.1. Characteristics of the Study Population

The main clinical characteristics of the study population are presented in Table 1. At hospital admission, 48 (22%), 72 (35%), and 204 (63%) patients had mild, moderate, and severe COVID-19, respectively.

### 3.2. Lipid Profile before COVID-19 and at Hospital Admission Due to COVID-19

Figure 2 shows lipid profile parameters in the entire study population before COVID-19 (from 1 month up to 1 year) and during COVID-19 (at hospital admission due to COVID-19). A significant reduction of TC, HDL-C, and LDL-C, (16%, 21%, and 18%, respectively) was observed between the two time points (*p* < 0.001 for all the comparisons). Instead, TG levels were stable over the examined time frame (*p* = 0.318). Significant covariates of Δ-TC were TSH (rho = 0.193, *p* = 0.001) and CRP (rho = *−*0.123, *p* = 0.031). Significant covariates of Δ-LDL-C were TSH (rho = 0.201, *p* = 0.001) and CRP (rho = *−*0.149, *p* = 0.010). Significant covariates of Δ-HDL-C were TSH (rho = 0.160, *p* = 0.008) and CRP (rho = *−*0.162, *p* = 0.005). Greater reductions of TC, LDL-C, and HDL-C were observed with increasing COVID-19 severity (*p* = 0.029, *p* = 0.006, and *p* = 0.008, respectively), while smaller reductions of TC, LDL-C, and HDL-C emerged in patients with previous anti-SARS-CoV-2 vaccination as compared to those without (*p* = 0.023, *p* = 0.005, and *p* = 0.003, respectively) as well as in those with high CCI (i.e., ≥7, the 75th percentile in the study population) as compared to those with low CCI (i.e., <7, the 75th percentile in the study population) (*p* = 0.003, *p* = 0.045, and *p* = 0.003, respectively). Moreover, Δ-HDL-C was lower in females as compared to males (*p* = 0.027), in patients with a history of previous cardiovascular (CV) events as compared to those without (*p* = 0.029), in patients with chronic kidney disease (CKD) as compared to those without (*p* = 0.017), and in those treated as compared to those untreated with statin therapy (*p* = 0.034).

### 3.3. Thyroid Function Test at Hospital Admission

In the entire study population, the median TSH level at hospital admission was 0.81 (0.45–1.68) μUI/mL. Forty-seven (14%) patients had low TSH levels (Table 2) and only two patients had high TSH levels. Among patients with low TSH levels, 9 had overt hyperthyroidism (i.e., low TSH, high FT3, and high FT4), 35 had NTIS (i.e., low TSH and low FT3 with variable FT4), and 3 had subclinical hyperthyroidism (i.e., low TSH, normal FT3, and normal FT4) (Table 2).

In the overall study population, TSH correlated with CRP (rho = *−*0.175, *p* = 0.004) (Figure 3). Moreover, higher TSH levels at hospital admission were found in patients with previous anti-SARS-CoV-2 vaccination as compared to those without (*p* = 0.027) (Figure 4), whereas lower TSH levels emerged with increasing COVID-19 severity (*p* for trend <0.001) (Figure 5).

### 3.4. Association between TSH and Lipid Changes between the Preinfection Period and Hospital Admission Due to COVID-19

In a linear regression analysis including Δ-TC as the dependent variable, and LG-TSH, LG-CRP, and COVID-19 severity as the independent variables, a significant association emerged between LG-TSH and Δ-TC (Table 3, Model 1A). The latter association remained significant after further adjusting for high CCI and anti-SARS-CoV-2 vaccination (Table 3, Model 1B). A significant association was found between LG-TSH and Δ-LDL-C in a linear regression model adjusted for LG-CRP and COVID-19 severity (Table 3, Model 2A) as well as in another linear regression model adjusted for LG-CRP, COVID-19 severity, high CCI, and anti-SARS-CoV-2 vaccination (Table 3, Model 2B). Instead, no significant association emerged between LG-TSH and Δ-HDL-C after adjusting either for sex, LG-CRP, and COVID-19 severity (Table 3, Model 3A) or sex, LG-CRP, COVID-19 severity, high CCI, and anti-SARS-CoV-2 vaccination (Table 3, Model 3B).

## 4. Discussion

To the best of our knowledge, this is the first study to assess the association between TSH levels and lipid metabolism parameters in patients with COVID-19. Indeed, some previous studies have explored either TSH levels or lipid profile in the acute phase of COVID-19, and only one study reported changes of lipid profile and TSH levels between the preinfection period and the acute phase of COVID-19 [18,29,30] without evaluating their association.

Three main results of the present study should be discussed: (1) the positive association between TC and LDL-C reduction between the preinfection period and hospitalization due to COVID-19 and either CRP levels or COVID-19 severity at hospital admission, (2) the negative association between TSH levels and either CRP levels or COVID-19 severity at hospital admission, and (3) the independent positive association between TSH levels at hospital admission and either TC or LDL-C reduction between the preinfection period and hospitalization due to COVID-19.

The observed positive association between TC and LDL-C reduction over the acute phase of COVID-19 and either CRP levels or COVID-19 severity at hospital admission is in line with the results of previous studies having shown lower TC and LDL-C levels in COVID-19 patients with a higher inflammatory burden and more severe clinical manifestations [8,31,32]. To date, several pathophysiological mechanisms have been proposed to explain the reduction of TC and LDL-C levels during the acute phase of COVID-19 in parallel with the increase of inflammatory biomarkers and disease severity. First, in analogy with what has been previously reported in other viral infections [33,34], it has been hypothesized that various proinflammatory cytokines produced during the acute phase of COVID-19 may negatively impact on hepatic lipogenesis [32]. Second, given that cholesterol is an essential structural component of enveloped viruses like SARS-CoV-2, it has been speculated that hypocholesterolemia may occur because of uncontrolled viral replication, a hallmark of the most severe clinical forms of COVID-19 [7,35]. Third, since liver dysfunction may be a clinical feature of severe COVID-19, lower TC and LDL-C levels have been hypothesized to result from reduced lipoprotein synthesis by dysfunctional hepatocytes in the most severe clinical forms of COVID-19 [29]. Fourth, as microcirculation is damaged and vascular permeability increases with increasing COVID-19 severity [36], it has been suggested that an abnormal distribution of lipoproteins from the intravascular to the extravascular space may contribute to reduce circulating TC and LDL-C in severe COVID-19 [19,29]. Of note, all these mechanisms are intriguing and biologically well-founded. Nonetheless, some of them need to be formally demonstrated. Moreover, it cannot be excluded that other biological pathways may have a pathophysiological role in lipid metabolism derangement during COVID-19, even independently from inflammation and disease severity.

The emerged negative association between TSH and either CRP levels or COVID-19 severity at hospital admission is consistent with pre-existing literature [12,17]. Overall, it suggests a dual perspective in which low TSH levels may be either a consequence or a risk factor of uncontrolled inflammatory response and disease severity in COVID-19. The first assumption (i.e., low TSH levels as a consequence of uncontrolled inflammatory response and disease severity) is in line with evidence reporting that a significant decrease of TSH levels may occur due to thyrotoxicosis and subclinical hyperthyroidism induced by subacute thyroiditis, one of the possible clinical complications of the most severe forms of COVID-19 [37,38]. Indeed, thyroid gland may be a target of both SARS-CoV-2 infection and SARS-CoV-2 infection-related inflammatory response [38]. Moreover, it agrees with evidence showing a higher incidence of nonthyroidal illness syndrome (NTIS) in COVID-19 patients with worse inflammatory profiles [13,39]. Indeed, inflammation may suppress the sensitivity of the hypothalamic–pituitary-axis to thyroid hormones levels promoting NTIS [40]. Consistently, hyperthyroidism, either clinically manifest or subclinical, and NTIS were the prevalent patterns of thyroid dysfunction in the present study. Instead, the second assumption (i.e., low TSH levels as a risk factor of uncontrolled inflammatory response and disease severity in COVID-19) is supported by some studies showing a possible detrimental impact of low TSH levels on immune response as well as an independent association between low TSH levels and worse clinical outcomes of COVID-19 [18,41]. Nonetheless, it should be emphasized that the two assumptions may be complementary, configuring a pathophysiological loop in which COVID-19 may trigger TSH decrease, which in turn may contribute to COVID-19 progression towards its most severe forms.

The main and unprecedented finding of this study is the independent association between lower TSH levels at hospital admission and the magnitude of TC and LDL-C reduction during COVID-19. From a pathophysiological perspective, it leads to hypothesize that lower TSH levels might contribute to lipid metabolism derangement in the acute phase of COVID-19, either indirectly, by reflecting higher levels of thyroid hormones, or directly. Due to the recognized association between hyperthyroidism (both overt and subclinical) and hypocholesterolemia [42], the first hypothesis (i.e., TSH indirectly contributes to lipid metabolism derangement during COVID-19) would be supported confirming the same association in patients with COVID-19. However, this latter association has not been explored by existing literature, yet. In addition, it cannot be verified in the present study. Indeed, among 47 COVID-19 patients with low TSH levels, only 12 had hyperthyroidism (9 had overt hyperthyroidism and 3 had subclinical hyperthyroidism). In addition, most of the patients (*n* = 275) had normal TSH levels with unknown levels of thyroid hormones. Thus, it was not possible to assess any difference in TC and LDL-C changes between COVID-19 patients with higher *versus* lower thyroid hormone levels in the presence of normal TSH levels. The second hypothesis (i.e., TSH directly contributes to lipid metabolism derangement during COVID-19) may be plausible based on different lines of evidence. Indeed, different observational studies have shown that in homeostatic conditions both TC and LDL-C levels increase proportionally with the increase of TSH levels and decline proportionally with the decline of TSH levels within the range of euthyroidism [43,44,45]. Moreover, the administration of recombinant human TSH to thyroidectomized patients has been reported to increase circulating TC levels [46]. Moreover, there is preclinical evidence showing that TSH receptors are present on the plasma membrane of hepatocytes and that upon TSH stimulation they can induce the expression of proprotein convertase subtilisin/kexin type 9 (PCSK9), leading to increased LDLR degradation and reduced hepatic uptake of LDL-C. In addition, it has been demonstrated that the activation of hepatic TSH receptors can induce the sterol regulatory element-binding protein 2 (SREBP2) signaling pathway, leading to a reduced conversion of cholesterol into bile acids within hepatocytes [42,47,48]. Therefore, irrespective of thyroid hormone levels, an increased TSH availability may reduce either LDLR-mediated LDL-C uptake or SREBP2-induced bile acid synthesis in hepatocytes, resulting in hypercholesterolemia, whereas a reduced TSH availability may result in hypocholesterolemia by promoting an increased hepatic expression of LDLR and activation of SREBP2.

Some limitations of the present study should be considered. First, the retrospective design of this study does not allow for ascertain any causal relationship between the study variables. Therefore, only speculative considerations can be drawn regarding the association between TSH levels at hospital admission and changes of lipid parameters. Second, upon hospital admission, a TSH reflex algorithm was routinely used for the assessment of thyroid function. Thus, it was not possible to detect levels of FT3 and FT4 in patients with normal TSH nor to evaluate the association between TSH and either TC or LDL-C reduction after correction for thyroid hormone levels. Third, TSH levels before hospital admission were not included among study variables as they were unavailable in medical records of most patients. This did not allow for correlation analyses between changes of TSH and changes of lipid parameters between the preinfection period and hospital admission.

## 5. Conclusions

In conclusion, the results of the present study confirm that hypocholesterolemia is a clinical feature of COVID-19 and suggest an intriguing pathophysiological connection between lower TSH levels, even in the range of euthyroidism, and cholesterol metabolism derangement during the acute phase of COVID-19. Based on literature data showing a significant association between either thyroid dysfunction or hypocholesterolemia and worse COVID-19 prognosis [8,9,10,17,18], it might be speculated that the coexistence of low TSH and cholesterol levels may further worsen the prognosis of patients hospitalized due to COVID-19. However, whether these combined metabolic abnormalities might represent a possible target for therapeutic intervention should be weighed against the debated evidence on this topic. Indeed, cholesterol-lowering therapy may display a favorable impact on COVID-19 outcomes [49,50]. Moreover, although it may be clinically useful to normalize TSH levels in the context of overt hyperthyroidism and in selected cases of subclinical hyperthyroidism [51], there is great controversy about treatment indication of NTIS, either with normal or low TSH levels [52]. Therefore, it is unlikely that promoting the restoration of preinfection cholesterol levels or correcting inappropriately low TSH levels may be clinically useful against COVID-19. Nonetheless, the possible existence of direct TSH-mediated effects on cholesterol metabolism is worthy of attention. Indeed, a better understanding of molecular pathways involved in TSH-mediated regulation of cholesterol metabolism may possibly pave the way for the identification of novel targets for cholesterol-lowering therapy.

## Figures and Tables

**Figure 1 jcm-11-03347-f001:**
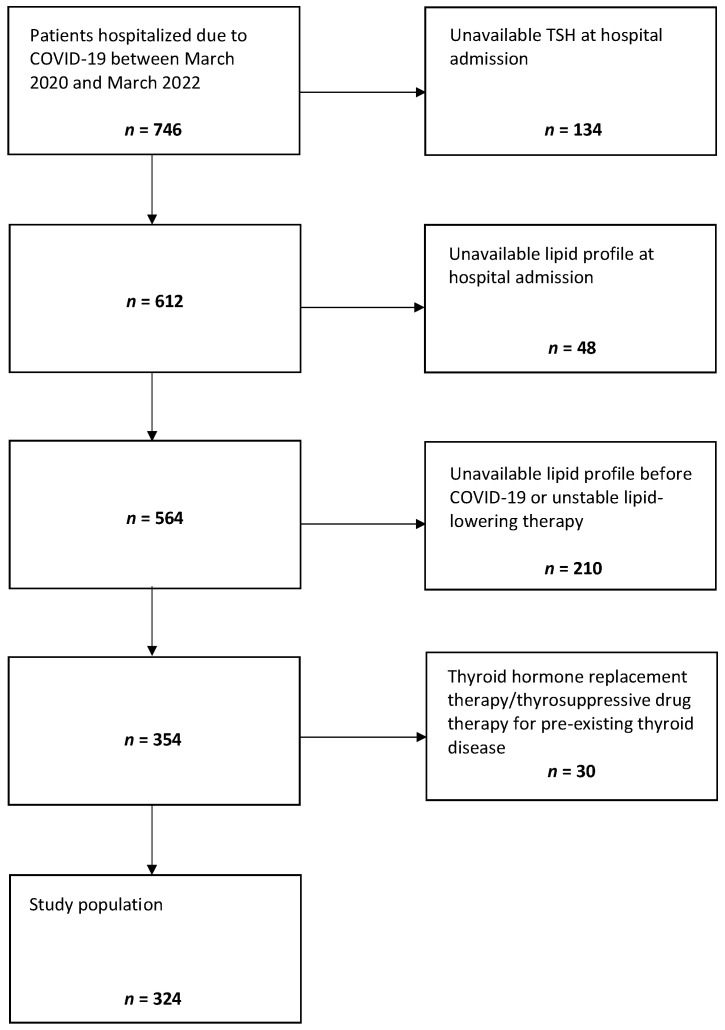
Selection of the study population.

**Figure 2 jcm-11-03347-f002:**
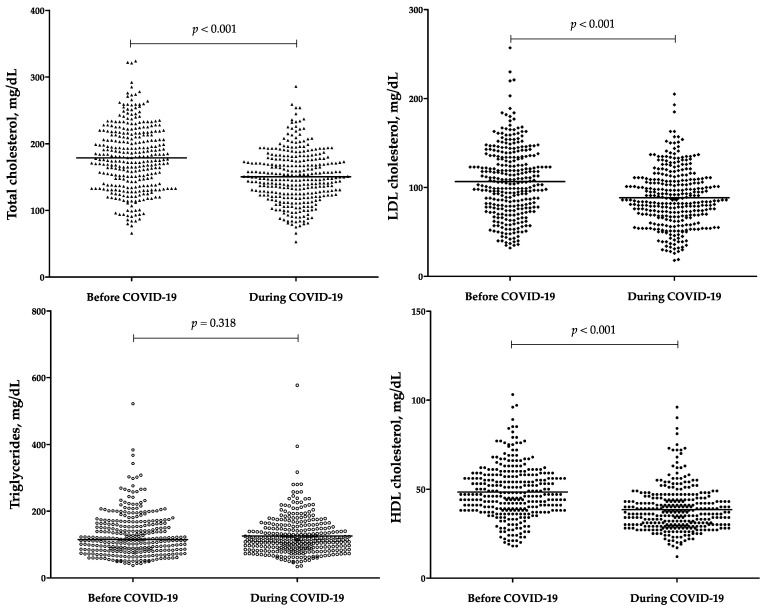
Lipid profile parameters before COVID-19 (from 1 month up to 1 year) and during COVID-19 (at hospital admission due to COVID-19). COVID-19, coronavirus disease 2019; HDL, high-density lipoprotein cholesterol; LDL, low-density lipoprotein cholesterol. Statistical significance was assessed through the paired samples t-test and the paired samples Wilcoxon test for parametric variables (i.e., total cholesterol, LDL cholesterol, HDL cholesterol) and nonparametric variables (i.e., triglycerides), respectively.

**Figure 3 jcm-11-03347-f003:**
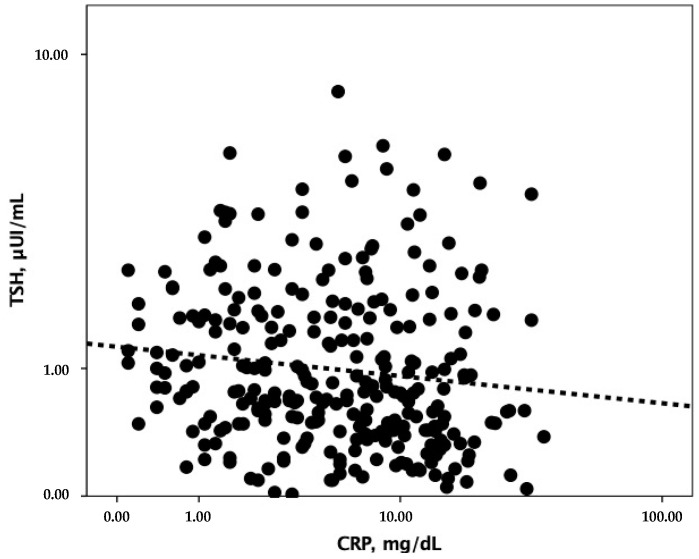
Correlation between TSH and CRP. CRP, C-reactive protein; TSH, thyroid stimulating hormone.

**Figure 4 jcm-11-03347-f004:**
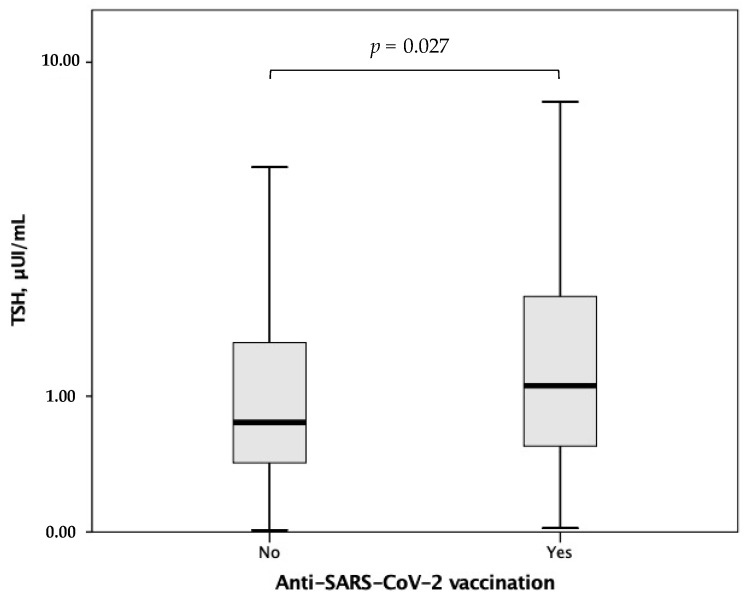
TSH levels according to anti-SARS-CoV-2 vaccination. TSH, thyroid-stimulating hormone. Statistical significance was assessed through the Mann–Whitney U-test.

**Figure 5 jcm-11-03347-f005:**
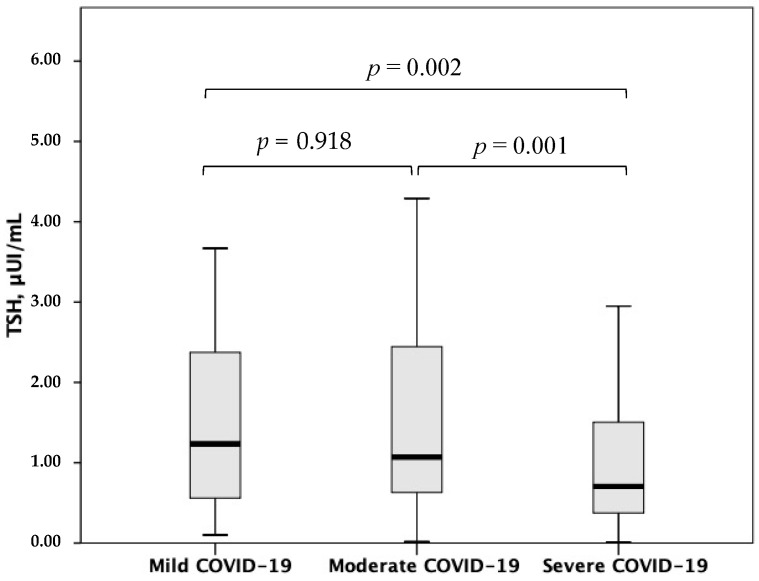
TSH levels according to COVID-19 severity. TSH, thyroid-stimulating hormone; COVID-19, coronavirus disease 2019. Statistical significance was assessed through the Mann–Whitney U-test.

**Table 1 jcm-11-03347-t001:** Characteristics of the study population.

	Study Population *n* = 324
**Age, years**	76 ± 15
**Male gender, %**	54
**BMI, kg/m^2^**	26 ± 4
**Current smoking, %**	11
**Hypertension, %**	67
**Type 2 diabetes, %**	24
**CKD, %**	18
**Previous CV event, %**	23
**Active cancer, %**	9
**Previous VTE, %**	43
**AF, %**	21
**Obesity, %**	24
**ACE inhibitors, %**	26
**ARBs, %**	14
**Other antihypertensive drugs, %**	31
**Oral anticoagulant therapy, %**	18
**Antiplatelet therapy, %**	29
**Oral hypoglycemic therapy, %**	10
**Insulin, %**	17
**Statins, %**	25
**Other lipid-lowering drugs, %**	2
**Previous anti-SARS-CoV-2 vaccination, %**	22
**Duration of symptoms before hospital admission, n of days**	5 (2–9)
**SBP, mmHg**	130 ± 19
**DBP, mmHg**	75 ± 10
**Leukocytes, ×10^3^/μL**	7 (5–10)
**Platelets, ×10^3^/μL**	197 (155–258)
**D-dimer, ng/mL**	933 (575–1917)
**Hs-cTn, ng/L**	17.8 (7.6–41.1)
**Glucose, mg/dL**	118 (98–154)
**eGFR, mL/min**	66 ± 28
**CRP, mg/dL**	5.7 (2.3–10.9)
**CCI**	5 (3–7)

Values are expressed as mean ± SD, median (25th–75th percentile), or percentage. ACE, angiotensin-converting enzyme; AF, atrial fibrillation; ARBs, angiotensin receptor blockers; BMI, body mass index; CCI, Charlson Comorbidity Index; CKD, chronic kidney disease; CRP, C-reactive protein; CV, cardiovascular; DBP, diastolic blood pressure; eGFR, estimated glomerular filtration rate; hs-cTn, high-sensitivity cardiac troponin; SBP, systolic blood pressure; VTE, venous thromboembolism.

**Table 2 jcm-11-03347-t002:** Distribution of thyroid hormone levels in patients with reduced TSH levels.

	Patients with Reduced TSH Levels *n* = 47
**↑** **FT3, n**	9
**↓** **FT3, n**	35
**↔** **FT3, n**	3
**↑** **FT4, n**	24
**↓** **FT4, n**	-
**↔** **FT4, n**	23
**↔** **FT3 and** **↔** **FT4, n**	3
**↔** **FT3 and** **↑** **FT4, n**	-
**↓** **FT3 and** **↔** **FT4, n**	20
**↓** **FT3 and** **↑** **FT4, n**	15
**↑** **FT3 and** **↔** **FT4, n**	-
**↑** **FT3 and** **↑** **FT4, n**	9

FT3, free triiodothyronine; FT4, free thyroxine; TSH, thyroid-stimulating hormone. ↑, high; ↓, low; ↔, normal.

**Table 3 jcm-11-03347-t003:** Association between TSH and either Δ-TC, Δ-LDL-C, or Δ-HDL-C.

**Dependent variable:** **Δ-TC**	**Model 1A**	**β**	** *p* **	**R square** **0.061** ***p* = 0.001**
	LG-TSH	0.150	**0.015**	
	LG-CRP	−0.108	0.094	
	COVID-19 severity	−0.092	0.155	
	**Model 1B**	**β**	** *p* **	**R square** **0.073** ***p* = 0.001**
	LG-TSH	0.125	**0.044**	
	LG-CRP	−0.115	0.079	
	COVID-19 severity	−0.093	0.181	
	High CCI	0.084	0.168	
	Anti-SARS-CoV-2 vaccination	0.048	0.468	
**Dependent variable:** **Δ-LDL-C**	**Model 2A**	**β**	** *p* **	**R square** **0.077** ***p* < 0.001**
	LG-TSH	0.153	**0.014**	
	LG-CRP	−0.097	0.138	
	COVID-19 severity	−0.141	**0.032**	
	**Model 2B**	**β**	** *p* **	**R square** **0.087** ***p* < 0.001**
	LG-TSH	0.131	**0.036**	
	LG-CRP	−0.101	0.134	
	COVID-19 severity	−0.130	0.066	
	High CCI	0.052	0.396	
	Anti-SARS-CoV-2 vaccination	0.068	0.314	
**Dependent variable:** **Δ-HDL-C**	**Model 3A**	**β**	** *p* **	**R square** **0.081** ***p* < 0.001**
	Male sex	0.160	**0.008**	
	LG-TSH	0.110	0.075	
	LG-CRP	−0.134	**0.041**	
	COVID-19 severity	−0.106	0.108	
	**Model 3B**	**β**	** *p* **	**R square** **0.118** ***p* < 0.001**
	Male sex	0.146	**0.017**	
	LG-TSH	0.076	0.216	
	LG-CRP	−0.131	**0.049**	
	COVID-19 severity	−0.124	0.074	
	High CCI	0.176	**0.004**	
	Anti-SARS-CoV-2 vaccination	0.038	0.575	

CCI, Charlson comorbidity index; COVID-19, coronavirus disease 2019; CRP, C-reactive protein; HDL-C, high-density lipoprotein cholesterol; LDL-C, low-density lipoprotein cholesterol; LG, logarithm; TC, total cholesterol; TSH, thyroid-stimulating hormone; Δ, variation.

## Data Availability

The data presented in this study are available on request from the corresponding author.

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
