# Peer review of "Thyroid-Stimulating Hormone Predicts Total Cholesterol and Low-Density Lipoprotein Cholesterol Reduction during the Acute Phase of COVID-19"

_jcm, 2022, doi:10.3390/jcm11123347_

Round 1

Reviewer 1 Report

This article focuses on the modification of the lipid profile of hospitalized patients due to SARS-Cov-2 infection, also bearing in mind the concentration of TSH as a predictive tool for these modifications.

Some aspects must be clarified to understand the conclusions of the study:

1. One of the most relevant aspects is the statistical analysis carried out and the interpretation of the results. Section 3.2 presents the correlations between different variables, all of them statistically significant considering the p value. The rho value determines the type of relationship between these variables. In all the data presented, this parameter is close to zero, so that, despite a significant correlation being observed by analysing the p value, there is no true linear relationship between the variables. It is likely that there is another type of correlation, but this is not linear. For this reason, it is recommended that this section be reviewed, since part of the study's conclusions derive from these observations.

2. On the other hand, a multiple linear regression analysis has been carried out to determine the association between TSH concentration and variations in the lipid profile, considering other variables that could be involved in this relationship. None of the models presented by the authors is valid, since no statistical significance is observed in all the variables analysed. Despite this, estimates are made of the possible relationships between the different variables. This is not correct when non-stable linear regression models are used. As in the previous case, it is recommended to review the proposed analysis to find more specific models with high statistical significance. It is recommended to analyse the heteroskedasticity, multicollinearity and the specification error before establishing the relationship models to determine the need to include new variables, detect out layers or the need to recode some variables if required.

Best regards

Author Response

We thank the Reviewer for this constructive comment which has given us the opportunity to further analyze our data in order to find the most suitable statistical approach to assess the study hypothesis. According to the Reviewer’s suggestion, we have checked the appropriateness of linear regression as the statistical analysis to model the association between either Δ-TC, Δ-LDL-C, or Δ-HDL-C and TSH as well as the appropriateness of two combinations of independent variables chosen among significant covariates of either Δ-TC, Δ-LDL-C, Δ-HDL-C, or TSH (i.e., CRP, COVID-19 severity, high CCI and anti-SARS-CoV-2 vaccination for models with Δ-TC and Δ-LDL-C as independent variables and sex, CRP, COVID-19 severity, high CCI and anti-SARS-CoV-2 vaccination for models with Δ-HDL-C as independent variable) by verifying 1) the linear association between each independent variable (if continuous) and the dependent variable, 2) the absence of multicollinearity between the independent variables, 3) the homoscedasticity, and 4) the normal distributions of residuals.

A curve fitting analysis was performed to examine the relationship between either Δ-TC, Δ-LDL-C or Δ-HDL-C and either TSH or CRP (the continuous variables among the putative explanatory variables). A significant linear relationship emerged between either Δ-TC, Δ-LDL-C or Δ-HDL-C and LG-TSH (R square=0.38 with p=0.001, R square=0.38 with p=0.001, R square=0.23 with p=0.012, respectively). Also, a significant linear relationship emerged between either Δ-TC, Δ-LDL-C or Δ-HDL-C and LG-CRP (R square=0.23 with p=0.008, R square=0.29 with p=0.004, and R square=0.14 with p=0.043, respectively).

In all the proposed models the independent variables had a variance inflation factor (VIF) <5, which conventionally excludes multicollinearity.

All the proposed models showed homoscedasticity at the scatterplots with the regression standardized predicted values on X-axis and the regression standardized residuals on the Y-axis.

All the proposed models showed a normal distribution of residuals at the Normal P-P plot.

Based on these analyses, the linear regression models proposed in the revised version of the manuscript seem reasonable as well as suitable to test the study hypothesis. Supporting this statement, it should be noted that, although in all the models R square values were quite low, p-values were significant, which suggests that all the models fitted data well albeit with limited explanatory power. Given that “the greater R-square the better the model” it cannot be excluded that other independent variables might possibly improve our ability to model the relationship between either Δ-TC, Δ-LDL-C or Δ-HDL-C and TSH. Nonetheless, in our data set no additional variables emerged to take into account as potential confounders for their clinically/statistically relevance.     

Overall, after such a deep revision of statistical analysis the main results of the study remained unchanged. Thus, we confirmed our discussion points, including interpretation of the results and conclusions.

Reviewer 2 Report

The authors describe a univariant and multivariant analyses of lipid panels and other data collected from patients before and after being hospitalized for Covid19.  While the manuscript is well written and findings are interesting, the tables and figures could be improved.

1)  Table 1.  The table should include information about the range of values for the study population where applicable.  The legend is incomplete, and it is unclear if the values listed represent the mean or median.  I'm guessing the values in parentheses are information about the distribution of values, but it is unclear why some only contain single values (STDV?) while others have a range of values (upper and lower limits?).  I think the table needs to be cleaned up and the legend expanded to include proper descriptions of all the data.

2)  The most important finding in the paper is presented in Table 2. but the presentation doesn't really do it justice.  In my opinion, this data would be easier to appreciate if the authors graphed them as box or violin plots. The individual data points should also be shown so that the reader can assess their distribution.  The figure legends should also include a description of how statistical significance was determined, Mann-Whitney or Student's t-test, equal or unequal variance, etc... 

3) Is there another way to present this data to highlight the categories of patients described in the text?  Overt hypothyroidism, etc.  I can't tell for sure but given how high the numbers are for patients with low FT3 and high FT4, I would expect a that a lot would have this combination?  Are these patients counted among those described as having variable FT4 in the text?  If FT4 is high while FT3 and TSH are low, would that suggest a problem with the response of peripheral tissues?  Also, I found the symbol used for variable to be a little confusing so it would be good to include a description in the legend, even if describing the up and down arrows seems a little silly.

4) The authors describe significant correlations between TSH levels and CRP and the severity of Covid19.  In the methods it says these results are based on linear regressions.  The authors should show plots of the data and the linear regressions with R-squared values and p-values so that the reader can fully appreciate the data.

5)  Table 4.  I can't think of a better way to show these results, but it is a lot to digest.  Perhaps the authors could put asterisks or make significant p-values red to help draw the reader's attention to significant findings.

Author Response

Reviewer:

The authors describe a univariant and multivariant analyses of lipid panels and other data collected from patients before and after being hospitalized for Covid19.  While the manuscript is well written and findings are interesting, the tables and figures could be improved.

Response:

We are thankful to the Reviewer for the positive comment.

Reviewer:

Table 1.  The table should include information about the range of values for the study population where applicable.  The legend is incomplete, and it is unclear if the values listed represent the mean or median.  I'm guessing the values in parentheses are information about the distribution of values, but it is unclear why some only contain single values (STDV?) while others have a range of values (upper and lower limits?). I think the table needs to be cleaned up and the legend expanded to include proper descriptions of all the data.

Response:

According to the Reviewer’s suggestion Table 1 has been revised to make explicit all the reported numerical values and their range [i.e., mean ± SD for parametric variables, median (25th-75th percentile) for non-parametric variables, or percentage for categorical variables]. Also, the Table legend has been expanded by including the description of data.

Reviewer:

The most important finding in the paper is presented in Table 2. but the presentation doesn't really do it justice.  In my opinion, this data would be easier to appreciate if the authors graphed them as box or violin plots. The individual data points should also be shown so that the reader can assess their distribution.  The figure legends should also include a description of how statistical significance was determined, Mann-Whitney or Student's t-test, equal or unequal variance, etc... 

Response:

According to the Reviewer comment, data shown in Table 2 have been plotted showing individual data points (see Figure 2). In the Figure legend the description of tests used for assessing statistical significance has also been included.

Reviewer:

Is there another way to present this data to highlight the categories of patients described in the text?  Overt hypothyroidism, etc. I can't tell for sure but given how high the numbers are for patients with low FT3 and high FT4, I would expect that a lot would have this combination?  Are these patients counted among those described as having variable FT4 in the text?  If FT4 is high while FT3 and TSH are low, would that suggest a problem with the response of peripheral tissues?  Also, I found the symbol used for variable to be a little confusing so it would be good to include a description in the legend, even if describing the up and down arrows seems a little silly.

Response:

We thank the Reviewer for this constructive comment. The revised version of Table 3 shows the distribution of different combinations of high/low/normal FT3 and FT4 levels as well as explanations for the symbols used for thyroid hormone levels. The condition characterized by low FT3 and high FT4 levels in the presence of low TSH levels was found in 15 out of 47 patients (a part of those 35 patients described as having variable FT4, the other part being represented by 20 patients with low FT3 and normal FT4 levels in the presence of low TSH levels). Regardless of FT4 levels, patients with low FT3 in the presence of low TSH levels were categorized as having NTIS (PMID: 24275187). Overall, there remains considerable debate regarding the precise mechanisms underpinning NTIS, with reported changes at all levels in the pathway of thyroid hormone synthesis/secretion, transport, cellular uptake and action. Nonetheless, both reduced Type 1 Iodothyronine Deiodinase (DIO1) and altered thyroid hormone receptor expression/signalling in the peripheral tissues have been described as possible pathogenic mechanisms, which could contribute to explain the presence of high FT4 levels in the presence of low FT3 and TSH levels in some cases (PMID: 24275187).

Reviewer:

The authors describe significant correlations between TSH levels and CRP and the severity of Covid19.  In the methods it says these results are based on linear regressions.  The authors should show plots of the data and the linear regressions with R-squared values and p-values so that the reader can fully appreciate the data.

Response:

According to the Reviewer’s suggestion in the revised version of the manuscript the crude association between TSH and either CRP, anti-SARS-CoV-2 vaccination, or COVID-19 severity has been graphically depicted (see Figure 3-5). In addition, R square and p-values have been made explicit for each model of linear regression (see the revised version of Table 4).

Reviewer:

Table 4.  I can't think of a better way to show these results, but it is a lot to digest. Perhaps the authors could put asterisks or make significant p-values red to help draw the reader's attention to significant findings.

Response:

According to the Reviewer’s suggestion, in the revised version of the table the most important results have been emphasized by putting significant p-values in bold.